# Dielectric Barrier Discharge Plasma Jet (DBDjet) Processed Reduced Graphene Oxide/Polypyrrole/Chitosan Nanocomposite Supercapacitors

**DOI:** 10.3390/polym13203585

**Published:** 2021-10-18

**Authors:** Chen Liu, Cheng-Wei Hung, I-Chung Cheng, Cheng-Che Hsu, I-Chun Cheng, Jian-Zhang Chen

**Affiliations:** 1Graduate Institute of Applied Mechanics, National Taiwan University, Taipei City 10617, Taiwan; R09543009@ntu.edu.tw; 2Advanced Research Center for Green Materials Science and Technology, National Taiwan University, Taipei City 10617, Taiwan; 3Department of Mechanical Engineering, National Taiwan University, Taipei City 10617, Taiwan; r08522744@ntu.edu.tw (C.-W.H.); ichungch@ntu.edu.tw (I.-C.C.); 4Department of Chemical Engineering, National Taiwan University, Taipei City 10617, Taiwan; chsu@ntu.edu.tw; 5Graduate Institute of Photonics and Optoelectronics, National Taiwan University, Taipei City 10617, Taiwan; iccheng@ntu.edu.tw; 6Department of Electrical Engineering, National Taiwan University, Taipei City 10617, Taiwan; 7Innovative Photonics Advanced Research Center (i-PARC), National Taiwan University, Taipei City 10617, Taiwan

**Keywords:** atmospheric-pressure plasma, dielectric barrier discharge, reduced graphene oxide, polypyrrole, supercapacitor, plasma

## Abstract

Reduced graphene oxide (rGO) and/or polypyrrole (PPy) are mixed with chitosan (CS) binder materials for screen-printing supercapacitors (SCs) on arc atmospheric-pressure plasma jet (APPJ)-treated carbon cloth. The performance of gel-electrolyte rGO/CS, PPy/CS, and rGO/PPy/CS SCs processed by a dielectric barrier discharge plasma jet (DBDjet) was assessed and compared. DBDjet processing improved the hydrophilicity of these three nanocomposite electrode materials. Electrochemical measurements including electrical impedance spectroscopy (EIS), cyclic voltammetry (CV), and galvanostatic charging-discharging (GCD) were used to evaluate the performance of the three types of SCs. The Trasatti method was used to evaluate the electric-double layer capacitance (EDLC) and pseudocapacitance (PC) of the capacitance. The energy and power density of the three types of SCs were illustrated and compared using Ragone plots. Our experiments verify that, with the same weight of active materials, the combined use of rGO and PPy in SCs can significantly increase the capacitance and improve the operation stability.

## 1. Introduction

Supercapacitors (SCs) have a higher capacitance and lower working voltage than regular capacitors; they are usually used in high-power-density applications requiring fast charging and discharging [1,2,3,4]. Their energy storage mechanisms are mainly the electrical double-layer capacitance (EDLC) and pseudocapacitance (PC). EDLC is manifested by rapid ion adsorption/desorption at the electrode/electrolyte interface [5]. This generally occurs in carbon-based electrode materials such as carbon black, carbon nanofibers, carbon nanotubes, and graphenes [6,7,8,9]. By contrast, PC is manifested by the Faraday oxidation/reduction reaction of active materials on the electrode surface [10,11,12]. Because graphene has high conductivity, carrier mobility, and specific surface area, it has been widely used in energy devices such as fuel cells [13], solar cells [14], oxygen evolution reaction electrodes [15], and SCs [16]. However, in practical applications, graphene may agglomerate and stack, and this may limit the charge accumulation in the electric double layer [17,18] and result in insufficient energy density. To further increase the capacitance, graphene is often compounded with conductive polymers to couple EDLC and PC [19]. Conducting polymers, such as polypyrrole (PPy), polyaniline, polythiophene, and their derivatives, are candidate active materials for realizing PCs owing to their good conductivity, excellent mechanical flexibility, high theoretical capacitance, and fast redox reaction properties [2,20,21,22]. Various studies have aimed to combine the advantages of SCs and traditional batteries to achieve high energy density and high power density. In this study, reduced graphene oxides (rGOs) and PPy were selected for the hybrid nanocomposite, and chitosan (CS) was used as a binder and dispersant to screen-print rGO/PPy/CS pastes onto carbon cloth.

Plasma has been extensively used for surface activation and modification [23,24,25]. Reactive plasma species may create active sites or defects, introduce dopants, and graft surface functional groups on materials surfaces [12,26,27]. All of these tailor the surface properties and may alter the surface wettability [1]. Plasma treatment is frequently performed to improve the quality of joining, bonding, or adhesion [28]. Low-pressure plasma operated with a vacuum system has been extensively used in industrial applications. Atmospheric-pressure plasma (APP) can be operated in a regular pressure environment without using a vacuum system; this is particularly advantageous for biomedicine and agriculture applications [29,30]. By avoiding the need for a vacuum system, the operation cost of APPs can be reduced, making APP an economical viable tool to perform plasma processing when the samples can tolerate environmental cleanliness conditions [4,27,31,32,33]. Owing to the vigorous interactions of APP reactive species on carbon-based materials, APPs have been extensively used for the rapid processing of carbonaceous materials [4,7,12,27,31,34,35].

In this study, we use a moderate temperature (~500 °C) nitrogen arc atmospheric pressure plasma jet (APPJ) to make a carbon cloth substrate hydrophilic. Because PPy cannot tolerate high temperature, near room temperature (<40 °C) He-2%O_2_ dielectric barrier discharge plasma jet (DBDjet) is used to treat the screen-printed rGO/PPy/CS for further improving its hydrophilicity, thereby facilitating its contact with the gel electrolyte. With the same material weight loading (i.e., the same weight of active materials), we compare the performance of SCs made with rGO/CS, PPy/CS, and rGO/PPy/CS. Our experimental results show improved capacitance and long-term operation stability when using rGO and PPy together as the active materials for SCs.

## 2. Materials and Methods

### 2.1. Preparation of rGO/PPy/CS, rGO/CS, and PPy/CS Pastes

CS (obtained from shrimp shells, deacetylation degree: >75%, Sigma Aldrich, Burlington, MA, USA) acetic acid solution was prepared by stirring a mixture of 0.3 g of CS powder and 20 mL of 0.1 M acetic acid (purity: >99.5%, AUECC, Kaohsiung, Taiwan) at 50 °C for 2 h. After natural cooling, the mixture was then stirred at room temperature for 1 h before use. Next, three types of active materials were introduced: 0.1 g of rGO (thickness: <5 nm, chip diameter: 0.1−5 μm, oxygen content: 5−10%, purity: >99%, Golden Innovation Business, New Taipei City, Taiwan), 0.1 g of PPy (conductivity: 10−50 S/cm, water: <1.0%, Sigma Aldrich, Burlington, MA, USA), and mixture of 0.05 g rGO and 0.05 g PPy. Then the powdered active material was mixed with 1.5 g ethanol (purity: 95%, Echo Chemical, Miaoli, Taiwan) and 3.6 g of CS acetic acid solution and stirred with a magnetic stirrer. Finally, the resulting mixture was concentrated by using a rotary evaporator at 50 °C for 150 s.

### 2.2. Fabrication of rGO/PPy/CS, rGO/CS, and PPy/CS Electrodes

A rectangular carbon cloth (3 cm × 4 cm) was used as the current collector. First, the carbon cloth was pre-treated using a nitrogen DC pulse APPJ (as shown in Figure 1) for 30 s. The treatment process is described in a previous study [1]. After the APPJ treatment, three types of active materials were screen printed on the carbon cloth with an area of 1.5 × 2 cm^2^. Next, the sample was calcined at 80 °C for 10 min in an oven. Finally, a DBDjet (as shown in Figure 1) was applied to post-treat the sample three times with a scanning speed of 2 mm/s. The carrier gas of the DBDjet was mixture with 98% helium and 2% oxygen.

### 2.3. Preparation of Gel-Electrolyte Solution

First, 1.5 g of polyvinyl alcohol (PVA; MW: 850,000−124,000, 99+% hydrolysis, Sigma Aldrich) powder was slowly added to 15 mL of 1 M sulfuric acid (purity: 95−97%, AUECC, Burlington, MA, USA). Then, the mixture was stirred at 200 rpm at 70 °C for 6 h. Then, the gel-electrolyte solution was naturally cooled and stirred at 300 rpm at room temperature.

### 2.4. Fabrication of Symmetric Sandwich-Type SCs

First, 1 mL of gel electrolyte solution was dropped on the DBDjet post-treated rGO/CS, PPy/CS, or rGO/PPy/CS nanocomposite-coated carbon cloth. The sample was then naturally dried for 24 h. The gel electrolyte coating step was repeated three times. Finally, the symmetrical sandwich-type SC was combined with two gel electrolyte-coated electrodes with the same active materials. Light pressing was applied against the sides of the gel electrolyte to assure the flatness of the device. Figure 1 shows a schematic of the process flowchart.

### 2.5. Characterization of rGO/PPy/CS, rGO/CS, and PPy/CS Nanocomposites and SCs

The water contact angles of the rGO/CS, PPy/CS, or rGO/PPy/CS nanocomposite-coated carbon cloths were measured by using a goniometer (Sindatek, Model 100SB, Taipei City, Taiwan). Scanning electron microscopy (SEM, JSM-IT100, JEOL, Tokyo, Japan) and X-ray photoelectron spectroscopy (XPS, Sigma Probe, Thermo VG-Scientific) were used to check the surface morphology and surface chemical bonding status of the rGO/PPy/CS, rGO/CS, and PPy/CS-coated carbon cloths. An electrochemical workstation (Autolab PGSTAT204, Metrohm, Utrecht, The Netherlands) was used to perform cyclic voltammetry (CV) (0−0.8 V, potential scan speed: 2−200 mV/s), galvanostatic charging discharging (GCD, 0−0.8 V, constant current: 0.25, 0.5, 1, 3, and 5 mA), and electrochemical impedance spectroscopy (EIS, 0.1–100,000 Hz) measurements in a two-electrode configuration to characterize the SC [1].

## 3. Results and Discussion

### 3.1. Surface Morphology of rGO/CS, PPy/CS, and rGO/PPy/CS Electrodes

Figure 2 shows the SEM images of the electrodes made with rGO/CS, PPy/CS, and rGO/PPy/CS pastes on carbon cloth after DBDjet post-treatment, respectively. The higher-magnification SEM images with rGO/CS paste reveal the sheet-like structure of rGO nano-flakes. rGOs with a large specific surface area typically provide EDLC for SCs [36]. The SEM images of the PPy/CS electrode show spherical PPy particles with coarse surfaces [37]. The higher-magnification SEM images with rGO/PPy/CS electrode show a combined structure, with a sheet-like structure of rGOs and spherical particles of PPy.

### 3.2. Hydrophilicity Assessment and XPS Results of rGO/CS, PPy/CS and rGO/PPy/CS Electrodes

Figure 3a shows that the water contact angle of pristine carbon cloth is 115.7°, indicating its hydrophobicity nature. After nitrogen arc APPJ pre-treatment, owning to the synergistic effect of high temperature and reactive plasma species, the carbon cloth became hydrophilic, and the droplet was immediately absorbed into the carbon cloth, as shown in Figure 3b [38]. Figure 3c-1,d-1,e-1 respectively shows the water contact angles of screen-printed rGO/CS, PPy/CS, and rGO/PPy/CS pastes on carbon clothes before DBDjet treatment. The PPy/CS-coated carbon cloth shows a water contact angle of 90.8° owing to the strong hydrophobicity of PPy. After DBDjet treatment on rGO/CS, PPy/CS, and rGO/PPy/CS-coated carbon clothes, the material surfaces became hydrophilic and absorbed water droplets, as shown in Figure 3c-2,d-2,e-2, respectively. The low temperature DBDjet processing can avoid the thermal damage on the coated materials (especially PPy) and simultaneously improve the wettability [1], thereby enhancing the interfacial contact with the follow-up-coated gel electrolytes.

XPS analyses were conducted to identify the chemical elements. The electrode containing rGOs showed higher C-C/C-O peak area ratios in XPS C1s spectra compared to that without rGOs. Further, the N1s spectra indicated the -NH^+^- content with the PPy coating, suggesting the successful deposition of PPy. Appendix A show the XPS results, and Appendix A show the corresponding element proportions (Appendix A).

### 3.3. EIS

Figure 4 shows the Nyquist plots of SCs with rGO/CS, PPy/CS, and rGO/PPy/CS nanocomposites. The inset of Figure 4 shows magnified view of the high frequency region. Table 1 lists the capacitive contribution of SCs with rGO/CS, PPy/CS, and rGO/PPy/CS nanocomposites. In the low frequency region, the curves are straight lines, corresponding to the Warburg diffusion impedance (W_0_). A straight line with a steeper slope indicates an electric double layer (EDL) with a compact structure. Conversely, one with a milder slope tends to indicate an ion diffusion in the capacitor. The SC with rGO/CS nanocomposites is more EDL-like; by contrast, that with PPy/CS nanocomposites is more ion-diffusion-like. Ideally, the SC with rGO/PPy/CS nanocomposites should exhibit a combination of these two effects. In the high-frequency region, a small recessed semicircle can be observed, indicating the parallel combination of the charge transfer resistance (R_CT_) and the constant phase element (CPE). The lower charge transfer resistance (indicated by a smaller semicircle) corresponds to a higher electron propagation speed. This is an important factor in fast redox systems, especially for SCs. However, among all SCs, there is no obvious difference in this regard. The difference of the Warburg diffusion impedance dominates the influence of each material on the SC. CPE corresponds to the interfacial capacitance between the electrode and the electrolyte. However, there is no obvious difference between each CPE value. Here, it should be noted that the intercept at the *x*-axis represents the series resistance (R_S_).

### 3.4. CV Measurement

Figure 5 shows the cyclic voltammetry (CV) measurement results of three different nanocomposite SCs at different scan rates. The enclosed area monotonically increased with the potential scan rate. The potential window and potential scan rate were set as 0–0.8 V and 2–200 mV/s, respectively. The areal capacitance can be calculated as [39]
(1)$$C_{A}\;=\;\frac{A}{s\;\times \;v\;\times \;\Delta V}$$

Here, *C_A_* is the areal capacitance (mF/cm^2^); *A*, the convolution area of the CV curves (mW); ∆*V*, the potential window (V); *v*, the potential scan rate (V/s); and *s*, the geometric area of the active part (cm^2^). An SC with rGO/CS nanocomposites mainly exhibits the EDLC, and therefore, the CV curve is more squarish [40]. By comparison, an SC with PPy/CS nanocomposites shows a much more irregular CV curve. This indicates the apparent oxidation–reduction reaction of PPy [41]. The SC with rGO/PPy/CS nanocomposites shows an obvious redox peak, and its areal capacitance (45.32 mF/cm^2^) is much higher than that of rGO/CS (23.37 mF/cm^2^) and PPy/CS (24.34 mF/cm^2^) at the scan rate of 2 mV/s. In this case, the areal capacitance of the SC with rGO/PPy/CS nanocomposite is boosted by both EDLC and PC. The combination of rGO and PPy dramatically improves the SC performance.

### 3.5. Trasatti’s Plots

Trasatti’s plots can be used to determine the contributions of the EDLC and the PC [42]. Trasatti’s theory is used to divide the storage mechanism into two different types: surface charge (C_out_) and diffusion control charge (C_in_). Different mechanism dominates in different potential scan rate ranges [43]. C_out_ is mainly formed by the accumulated charge on the active material surface, which is the origin of EDLC. Conversely, the charges stored in the PC-based material contribute to C_in_. By extrapolation, the two types of capacitance values can be evaluated. When the voltage scan rate approaches infinity, the C_out_ dominates the capacitance value. When the voltage scan rate approaches zero, there is enough reaction time for the charge to diffuse into the active material, and charges can be stored internally and externally (C_total_ = C_in_ + C_out_) [44,45].

Figure 6 shows Trasatti’s plots of SCs with rGO/CS, PPy/CS, and rGO/PPy/CS nanocomposites. Figure 6a shows the relationship between *1/C_A_* and *V*^0.5^; this straight-fitted line can be used to extrapolate C_total_. Figure 6b shows the relationship between *C_A_* and *v*^−0.5^; it can be used to extrapolate C_out_. Table 2 shows the calculated capacitance contribution. The SCs with rGO/CS nanocomposites exhibit higher EDLC (79.4%), whereas those with PPy/CS nanocomposites exhibit roughly equal contributions of EDLC and PC (50%:50%). The rGOs added into PPy/CS to form rGO/PPy/CS boost the EDLC effect to achieve the best performance with the corresponding capacitive contribution percentage of EDLC and PC being ~75%:25%.

### 3.6. GCD

Figure 7 shows the GCD results of the SCs with rGO/PPy/CS, rGO/CS, and PPy/CS nanocomposites. GCD measurements can be used to calculate the areal capacitance values as [46]
(2)$$C_{A}\;=\;\frac{2I\;\times \;T}{s\;\times \;\Delta V}$$

Here, *C_A_* is the areal capacitance (mF/cm^2^); *I*, the discharging current (mA); *T*, the discharging time (*s*); ∆*V*, the potential window (*V*); and s, the geometric area of the active material (cm^2^). Table 3 lists the areal capacitance values calculated from the GCD curves in Figure 7. The SC with rGO/PPy/CS nanocomposites shows the best performance with an areal capacitance of 72.79 mF/cm^2^ under a discharging current of 1 mA; this is 2–3 times higher than that achieved under the other two conditions. This result agrees with the CV results. With the PPy/CS composite, the slope of the GCD curve changes for a low discharging current. This phenomenon indicates that PPy performs a more obvious redox reaction [1]. The parameter used to compare the performance of SCs is gravimetric-specific capacitance. In our case, however, the measurement of weight change due to screen-printed materials in our SCs could lead to large error in determining gravimetric-specific capacitance because of the materials loading, and therefore, we used areal capacitance to compare our results. In addition, the areal capacitance and/or volumetric capacitance is more useful to describe the specification in commercial SCs. Appendix A lists the C_A_ performance of various composite SCs. The *C_A_* value of SC with rGO/PPy/CS nanocomposite is on par with most of the results in Appendix A.

### 3.7. Ragone Plots

Ragone plots are used to evaluate the energy and power density of the SCs. The areal energy density and areal power density can respectively be calculated as [47]
(3)$$E_{A}\;=\;\frac{C_{A}\;\times \;\Delta V^{2}}{7.2}$$
(4)$$P_{A}\;=\;\frac{3.6\;\times \;E_{A}}{T}$$
where *EA* is the energy density (µWh/cm^2^); *C_A_*, the areal capacitance from the GCD measurement (mF/cm^2^); ∆*V*, the potential window (*V*); *PA*, the power density (mW/cm^2^); and *T*, the discharging time (*s*).

Figure 8 shows Ragone plots of the SCs with rGO/CS, PPy/CS, and rGO/PPy/CS nanocomposites. The SC with rGO/PPy/CS nanocomposites exhibited the best achieved energy density of 6.5 µWh/cm^2^ under a discharging current of 0.25 mA and the best achieved power density of 1.333 mW/cm^2^ under a discharging current of 5 mA. The nanocomposite materials increased the energy density. By contrast, the SCs with rGO/CS and PPy/CS nanocomposites showed poor performance, indicating the advantages of mixing rGOs, PPy, and CS to form nanocomposites.

### 3.8. CV Stability and Bending Tests

Figure 9a shows the CV stability test results of the SCs with a potential scanning rate of 200 mV/s. After 10,000 cycles, rGO/CS SC retains 90% of the initial areal capacitance, whereas PPy/CS retains only 43% owing to the poor PPy stability in long-term operation. rGO/PPy/CS SC retains 88% of the initial capacitance value. Adding rGOs to PPy/CS clearly improves the CV stability.

Figure 9b shows the bending test results of the SCs under different curvatures. The capacitance values are measured by GCD under the discharging current of 1 mA. The capacitance values fluctuate randomly as the curvature increases. Interestingly, the capacitance values increase at a certain curvature. All SCs function well under bending.

## 4. Conclusions

This study evaluates the screen-printed rGO/CS, PPy/CS, and rGO/PPy/CS SCs with active materials of the same weight. The GCD results show that the areal capacitance of rGO/CS, PPy/CS, and rGO/PPy/CS SCs is 26.23, 46.34, and 72.79 mF/cm^2^, respectively. Combining rGO and PPy clearly results in improved areal capacitance and improved 10,000-cycle CV stability. The EIS results indicate that the SC with rGO/CS nanocomposites is more EDL-like, whereas the one with PPy/CS nanocomposites is more ion-diffusion-like. Trasatti analysis indicates that the addition of rGOs increases the contribution of EDLC. All three types of SCs perform well under bending.

## Figures and Tables

**Figure 1 polymers-13-03585-f001:**
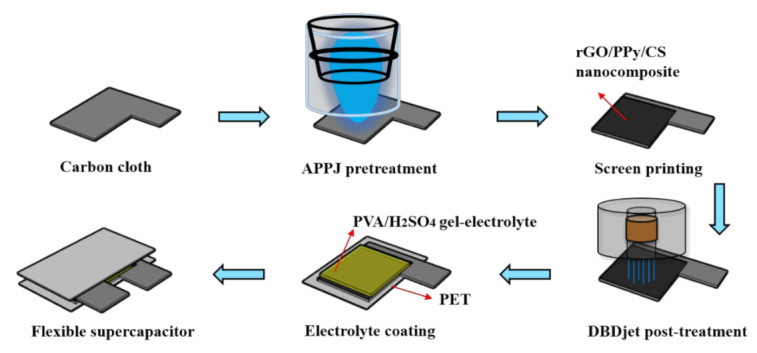
Schematic diagram of fabrication process of flexible SCs.

**Figure 2 polymers-13-03585-f002:**
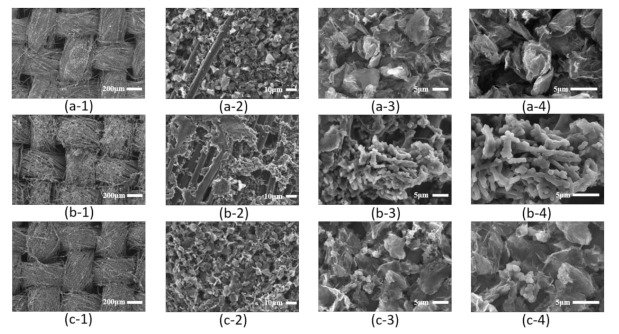
SEM images. rGO/CS electrodes with (**a-1**) 70×, (**a-2**) 1000×, (**a-3**) 3000×, and (**a-4**) 5000× magnification; PPy/CS electrodes with (**b-1**) 70×, (**b-2**) 1000×, (**b-3**) 3000×, and (**b-4**) 5000× magnification; and rGO/PPy/CS electrodes with (**c-1**) 70×, (**c-2**) 1000×, (**c-3**) 3000×, and (**c-4**) 5000× magnification.

**Figure 3 polymers-13-03585-f003:**
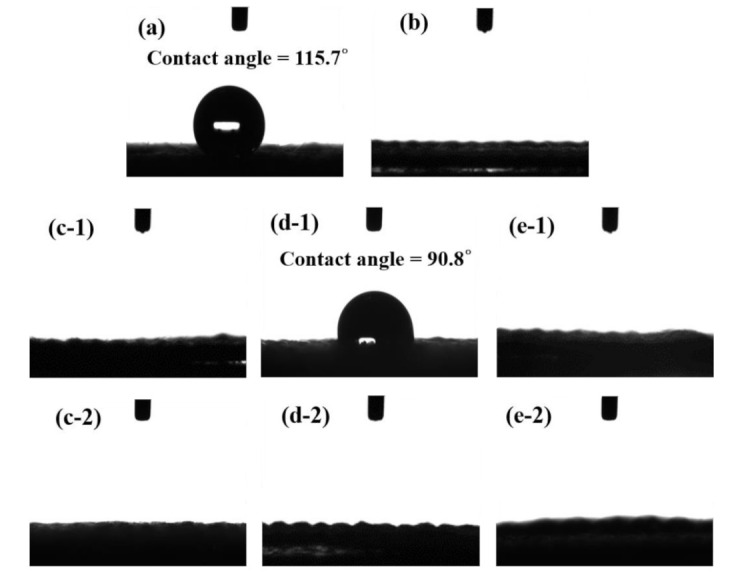
Water contact angle of pure carbon cloth (**a**) without APPJ treatment and (**b**) with APPJ treatment; rGO/CS-coated carbon cloth (**c-1**) without DBDjet treatment and (**c-2**) with DBDjet treatment; PPy/CS-coated carbon cloth (**d-1**) without DBDjet treatment and (**d-2**) with DBDjet treatment; and rGO/PPy/CS-coated carbon cloth (**e-1**) without DBDjet treatment and (**e-2**) with DBDjet treatment.

**Figure 4 polymers-13-03585-f004:**
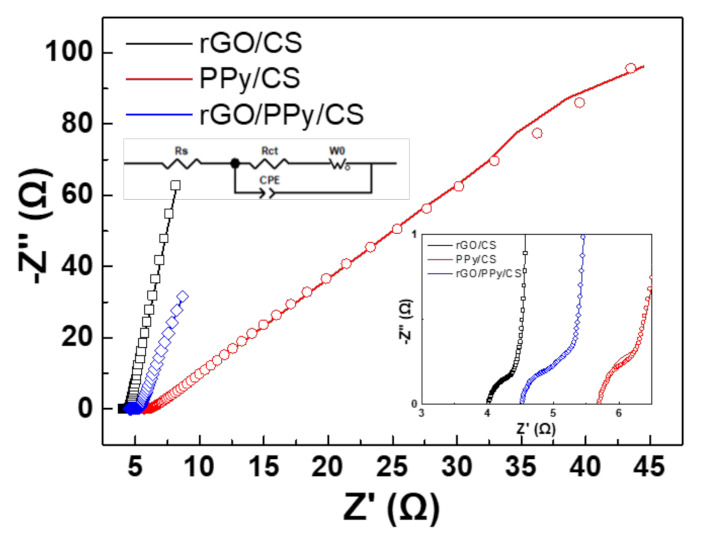
Nyquist plots of SCs with rGO/CS, PPy/CS, and rGO/PPy/CS nanocomposites.

**Figure 5 polymers-13-03585-f005:**
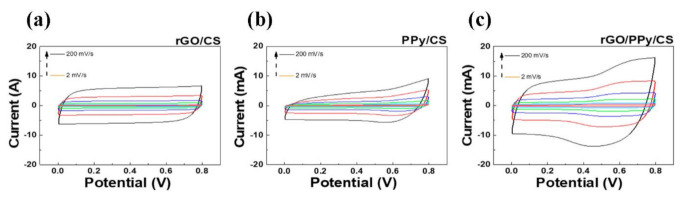
Cyclic voltammetry curves of (**a**) rGO/CS, (**b**) PPy/CS, (**c**) rGO/PPy/CS nanocomposite SCs at different scan rates.

**Figure 6 polymers-13-03585-f006:**
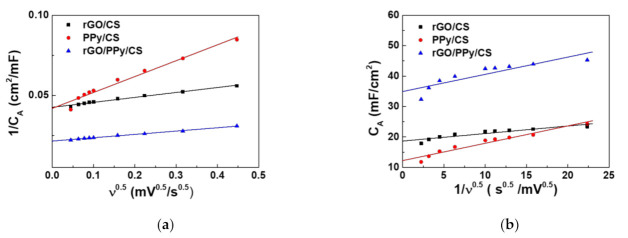
Trasatti’s plots of SCs with rGO/PPy/CS, rGO/CS, and PPy/CS nanocomposites: (**a**) 1/C_A_ vs. *v*^1/2^. (**b**) CA vs. 1/*v*^1/2^.

**Figure 7 polymers-13-03585-f007:**
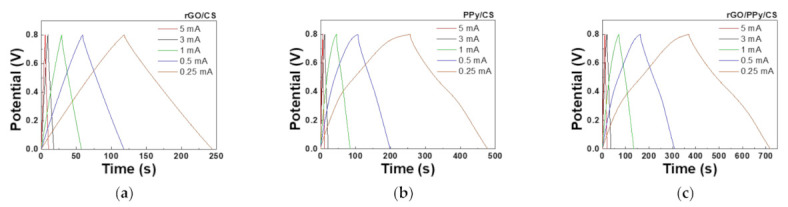
GCD curves of the SCs with (**a**) rGO/CS, (**b**) PPy/CS, and (**c**) rGO/PPy/CS nanocomposites under different discharge currents.

**Figure 8 polymers-13-03585-f008:**
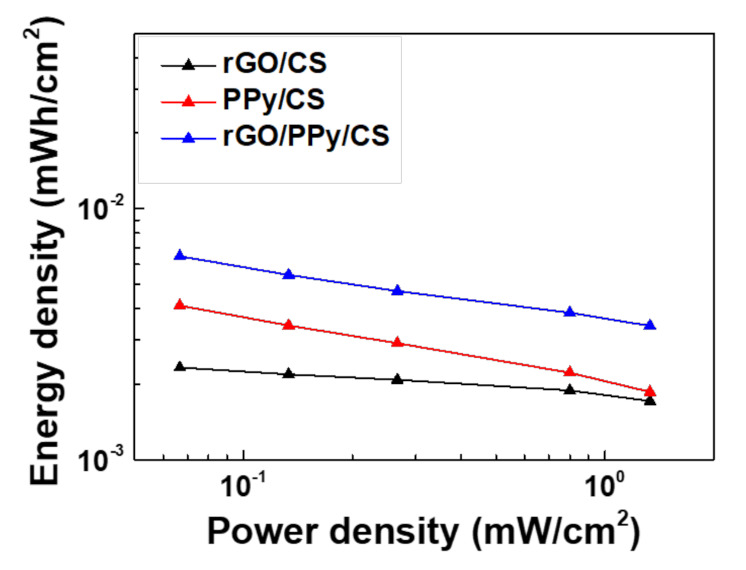
Ragone’s plots of SCs with rGO/CS, PPy/CS and rGO/PPy/CS nanocomposite.

**Figure 9 polymers-13-03585-f009:**
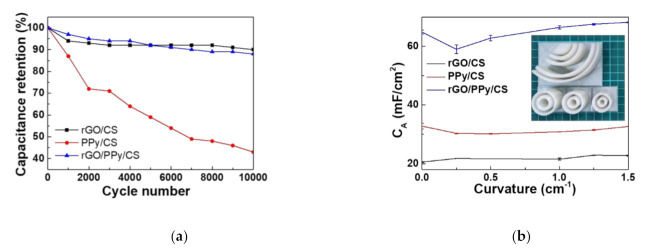
(**a**) Cycling stability test with a potential scan rate of 200 mV/s. (**b**) Bending test under different curvatures with the discharging current of 1 mA.

**Table 1 polymers-13-03585-t001:** Capacitive contribution of SCs with rGO/CS, PPy/CS, and rGO/PPy/CS nanocomposites.

	R_S_ (Ω)	R_CT_ (Ω)	W_0_ (Ω)	C_PE_ (µs^α^/Ω)
rGO/CS	4.05	0.16	1.0	0.0002
PPy/CS	5.72	0.72	704.7	0.0015
rGO/PPy/CS	4.53	0.45	1.3	0.002

**Table 2 polymers-13-03585-t002:** Capacitive contributions of SCs with rGO/CS, PPy/CS, and rGO/PPy/CS nanocomposites.

	Ctotal (mF/cm^2^)	Cin (mF/cm^2^)	Cout (mF/cm^2^)	Capacitive Contribution (EDLC:PC) (%)
rGO/CS	23.5	18.7	4.9	79.4:20.6
PPy/CS	23.9	12.2	11.6	51.3:48.7
rGO/PPy/Cs	46.6	34.9	11.7	75.0:25.0

**Table 3 polymers-13-03585-t003:** Areal capacitance of SCs analyzed from GCD measurement.

Areal Capacitance (mF/cm^2^)					
	Discharging Current (mA)				
	5	3	1	0.5	0.25
rGO/CS	19.30	21.25	23.42	24.64	26.23
PPy/CS	20.97	25.01	32.77	38.57	46.34
rGO/PPy/Cs	38.44	43.32	52.91	61.33	72.79

## Data Availability

The data presented in this study are available on request from the corresponding author.

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
