# Peer review of "Dielectric Barrier Discharge Plasma Jet (DBDjet) Processed Reduced Graphene Oxide/Polypyrrole/Chitosan Nanocomposite Supercapacitors"

_polymers, 2021, doi:10.3390/polym13203585_

Round 1
Reviewer 1 Report
The manuscript entitled “Dielectric barrier discharge plasma jet (DBDjet) processed reduced graphene oxide/polypyrrole/chitosan nanocomposite supercapacitors“ by C. Liu et al. reports new results concerning the application of composites based on reduced graphene oxide (rGO), polypyrrole (PPY) and chitosan in the screen-printing supercapacitors field. The discussions are well structured and experimental results prove conclusions of this work. In order to improve this manuscript, I suggest to authors to add after section 3.1, a characterization of composites by Raman scattering and IR spectroscopy. A deep analysis of the Raman, IR and XPS spectra is necessary to be included in the revised manuscript. At the end of this manuscript, a comment concerning the performance of supercapacitors based on rGO, PPY and chitosan in comparison with other composite materials will be included.
I recommend this article to be published in the Polymers journal after the major revision.
Author Response
We thank the reviewers for their valuable comments. We have revised the paper accordingly. Point-to-point responses are listed below. The revision parts are highlighted in red in the revised manuscript.
Reviewer 1
The manuscript entitled “Dielectric barrier discharge plasma jet (DBDjet) processed reduced graphene oxide/polypyrrole/chitosan nanocomposite supercapacitors“ by C. Liu et al. reports new results concerning the application of composites based on reduced graphene oxide (rGO), polypyrrole (PPY) and chitosan in the screen-printing supercapacitors field. The discussions are well structured and experimental results prove conclusions of this work. In order to improve this manuscript, I suggest to authors to add after section 3.1, a characterization of composites by Raman scattering and IR spectroscopy. A deep analysis of the Raman, IR and XPS spectra is necessary to be included in the revised manuscript. At the end of this manuscript, a comment concerning the performance of supercapacitors based on rGO, PPY and chitosan in comparison with other composite materials will be included.
I recommend this article to be published in the Polymers journal after the major revision.
Response) We thank the reviewer for the valuable comments. We have added a table to compare the properties of our PPy-rGO-CS supercapacitors with those from literatures in Table S5 in the Supplementary Information. In our manuscript, we have already included XPS analyses. However, as for Raman and IR analyses, we completely agree with the reviewer that these analyses are very important for graphene material system, especially for those with mechanically exfoliated-and-transferred graphene and CVD-grown graphene because graphene fabricated by these methods is much less defective. As for our materials prepared with our screen-printed pastes, we have worked on this kind of screen-printed materials for several years (since 2013), including bare rGO, bare CNT, rGO-CNT, rGO-carbon black, rGO-Pt, rGO-ZnO, rGO-SnO2, rGO-PANI, rGO-PPy. We used low cost screen printing method to fabricate flexible supercapacitor on carbon cloth. In fact, the materials produced using this method is quite defective and highly irregular in morphology. Carbon cloth itself are weaved fibers and is not flat. After screen-printing these materials, the surface is highly irregular. We have tried Raman measurement in our previous system and we found the signals are quite inconsistent. As for IR measurement, because carbon cloth is used as substrate with materials screen-printed on it, the only possible way to perform IR measurement is in reflective IR mode. However, carbon cloth itself has extruded fibers with irregular screen printed materials. The reflective signals are highly improbable to detect. As such, we didn’t perform Raman and IR analyses.
Reviewer 2
The manuscript of Liu et al. describes the development of supercapacitors based on reduced graphene oxide/polypyrrole/chitosan nanocomposite supercapacitors. It is very important to situate the reader in the frontier of knowledge. Based on this aspect, it is important to highlight the novelty and the advantages of this experimental system. Please compare your experimental system with data available in the literature (a new table and the direct inspection of points in the Ragone plot are also necessary).
Response) We thank the reviewer for the excellent comments. We have revised the paper accordingly to improve our manuscript. We have added a table (Table S5) in the supplementary information to compare the results with those in literatures. Also, we highlight the goal of this study in the introduction section in the revised manuscript.
Line 99: It is “5 mL” and not “5 ml”
Response) We thank the reviewer to point this out. We have revised it.
Figure 2 – it is not possible to visualize the magnification bar – please improve the quality of the Figure.
Response) We have revised the photos accordingly.
Figure 5 – It would be interesting to provide curves at different scan rates
Response) We have added CV curves with different scan rates.
Figure 8 – I suggest performing the Ragone plot with 5 points
Response) We have performed more measurement to increase the number of data points to five.
Figure 9 – Please include the bar errors in all of the results.
Response) For SCs bended under different curvatures, we have added error bar with 5 measurements. As for 10000-cycle cycling stability test, it is a long-time test, we show the representative data points.
Based on these aspects, I consider that major revisions are required for this manuscript.

Reviewer 2 Report
The manuscript of Liu et al. describes the development of supercapacitors based on reduced graphene oxide/polypyrrole/chitosan nanocomposite supercapacitors. It is very important to situate the reader in the frontier of knowledge. Based on this aspect, it is important to highlight the novelty and the advantages of this experimental system. Please compare your experimental system with data available in the literature (a new table and the direct inspection of points in the Ragone plot are also necessary).
Line 99: It is “5 mL” and not “5 ml”
Figure 2 – it is not possible to visualize the magnification bar – please improve the quality of the Figure.
Figure 5 – It would be interesting to provide curves at different scan rates
Figure 8 – I suggest performing the Ragone plot with 5 points
Figure 9 – Please include the bar errors in all of the results.
Based on these aspects, I consider that major revisions are required for this manuscript.
Author Response

(The authors gave the same response as above.)

Round 2
Reviewer 1 Report
I recommend this article to be published in the Polymers journal in the present form.
Reviewer 2 Report
Based on modifications provided by the authors, the manuscript is ready to be accepted for publication.